# Characterization of the hemodynamic response function in white matter tracts for event-related fMRI

Muwei Li [1,2], Allen T. Newton[1,2], Adam W. Anderson[1,2,3], Zhaohua Ding[1,3,4] & John C. Gore[1,2,3]

Accurate estimates of the BOLD hemodynamic response function (HRF) are crucial for the interpretation and analysis of event-related functional MRI data. To date, however, there have been no comprehensive measurements of the HRF in white matter (WM) despite increasing evidence that BOLD signals in WM change after a stimulus. We performed an event-related cognitive task (Stroop color-word interference) to measure the HRF in selected human WM pathways. The task was chosen in order to produce robust, distributed centers of activity throughout the cortex. To measure the HRF in WM, fiber tracts were reconstructed between each pair of activated cortical areas. We observed clear task-specific HRFs with reduced magnitudes, delayed onsets and prolonged initial dips in WM tracts compared with activated grey matter, thus calling for significant changes to current standard models for accurately characterizing the HRFs in WM and for modifications of standard methods of analysis of functional imaging data.

[1] Vanderbilt University Institute of Imaging Science, 1161 21st Ave. S, Medical Center North, AA-1105, Nashville, TN 37232, USA. [2] Department of Radiology and Radiological Sciences, Vanderbilt University Medical Center, 1161 21st Ave. S, Medical Center North, Nashville, TN 37232, USA. [3] Department of Biomedical Engineering, Vanderbilt University, 2301 Vanderbilt Place, Nashville, TN 37235, USA. [4] Department of Electrical Engineering and Computer Science, Vanderbilt University, 2301 Vanderbilt Place, Nashville, TN 37235, USA. Correspondence and requests for materials should be addressed to Z.D. (email: zhaohua.ding@vanderbilt.edu) or to J.C.G. (email: john.gore@vumc.org)

Magnetic resonance imaging (MRI) is well established as a means to delineate both functional and anatomic circuits in the brain using functional and diffusion MRI, respectively. Distant grey matter (GM) cortices containing neuron cell bodies may be engaged together in specific functions and are connected by bundles of axons within white matter (WM). Active GM volumes require higher rates of oxygen delivery and blood flow, which then cause increased blood oxygenation-level dependent (BOLD) signals that are detectable by functional magnetic resonance imaging (fMRI)[1]. The axon bundles (or white matter (WM) tracts) can be identified using diffusion tensor imaging (DTI) or similar diffusion methods which exploit the anisotropy of water displacements in WM fibers[2]. These two modalities (fMRI and DTI) thus provide a macroscopic description of neural networks in terms of how distinct GM areas interactively respond to a specific functional task and how these areas are anatomically connected[3,4]. However, conventional DTI does not reveal dynamic changes within the WM tracts and does not report on function-related signals that may be present in WM.

In recent years, a growing number of studies have reliably detected BOLD signals within WM which appear to reflect intrinsic neural activity[5–9]. For example, Mazerolle et al. observed WM activation in the corpus callosum while performing the inter-hemispheric transfer task[10]. Gawryluk et al. detected WM activations distributed in distinct callosal areas in response to the Sperry and Poffenberger tasks, respectively[11]. However, the regions activated were relatively small and in general the sensitivity for detecting WM activity has been much lower than in GM. While much of the explanation for this may be due to the lower vascular volume of WM, the sensitivity for detection is also reduced whenever incorrect assumptions about the time course of responses are incorporated into data analyses[12]. Conventionally, general linear models (GLM) assume that BOLD responses are derived by convolution with a canonical hemodynamic response function, measurable as the transient signal change arising after a short stimulus. Incorrect selections of the HRF can dramatically reduce the number and magnitudes of BOLD activations. From previous studies, delayed BOLD responses are commonly observed in WM[13–15]. Using a set of optimized HRFs with successive one-second time shifts, WM activity in the corpus callosum was detected with increased sensitivity by Courtemanche et al.[16]. A key finding from this work was that WM activations were highly dependent on the timing of the reference HRF. Similarly, Tae et al. studied neural activity in the corpus callosum using acoustic stimulation with the reference HRF artificially shifted by one or two TR values[17] and found that activation of the corpus callosum could be more effectively identified with time-shifted HRF models. Though promising, these studies were based on a simple time shifting of the canonical HRFs for GM. For more robust detection of WM activation, explicit characterizations of WM-specific HRFs are highly desired.

These previous studies were performed using block-design tasks, consisting of several alternating epochs of "on-off" periods throughout a run, which offer the potential of increasing the detection power. However, many cognitive paradigms make use of event-related acquisitions which avoid effects such as habituation and allow specific transient brain responses to be recorded e.g. in "oddball" studies[18,19]. When individual stimuli are short-lived and well spaced, the BOLD signal produced replicates the HRF and allows measurement of the dynamic response of cortical areas. Simple sensory stimuli evoke HRF responses in relatively confined regions of cortex. We aimed to measure the explicit form of the HRF in several WM tracts simultaneously. For this purpose, we used an event-related cognitive task that is known to engage multiple, separate cortical regions that each display an HRF. By this means we aimed to detect the transient signals in WM tracts connecting different parts of the brain. We adopted an event-related cognitive paradigm based on the Stroop word-color task[20] which we have previously shown produces robust transient BOLD signals in different GM regions[21].

Our previous studies demonstrated that functional activity can be detected with increased sensitivity by segmenting specific WM tracts and integrating their BOLD signals[6,7], so here the HRF was analyzed on a tract basis. First, a number of activated GM clusters were identified using a standard GLM incorporating the GM-specific HRF as a kernel. Second, WM tracts, where reliably identifiable, were reconstructed between each pair of activated clusters by a DTI tractography approach. Finally, the BOLD time courses within these specific WM tracts were integrated, parameterized and evaluated. Transient BOLD changes are detectable in task-specific WM tracts in event-related fMRI, and the HRFs in WM have smaller magnitudes, increased times to peak and prolonged initial dips compared to those in activated GM. The current study provides data for quantitative modeling of BOLD responses in WM, from which physiological processes and neurovascular coupling in WM may be better understood.

## Results

**Brain activation detected using canonical HRF for GM.** The experimental paradigm (see Methods for details) of the Stroop word-color task is shown in Fig. 1a. Twenty subjects completed the Stroop word-color test with an accuracy of 96.5 ± 3.1% (mean ± standard deviation over twenty subjects) and a response time of 724.4 ± 109.8 ms for congruent events, and with an accuracy of 90.3 ± 9.1% and a response time of 937.3 ± 202.9 ms for incongruent events, consistent with the conflict between the task of color naming and automatic reading. As shown in Fig. 1b, seven major activated GM clusters were detected ($P < 0.05$, two sample $t$-test, FWE corrected) by contrasting the incongruent events against congruent events across the population. Each cluster was identified by a name that represented the anatomical structure it overlapped with the highest probability (See Table 1 for a summary of the names, coordinates and overlapped structures). Among 21 possible routes connecting the seven GM clusters, 11 WM tracts (Fig. 2) were reproducibly reconstructed by DTI tractography across all the subjects, as shown in Supplementary Table 1. Other tracts were reliably segmented in only a subset of subjects.

**Distinct HRF profiles in WM compared to GM.** The event-related signal time course (impulse response $h$) was obtained by a deconvolution approach to the measured fMRI time series (output signal $y$) based on the known incongruent stimuli (input signal $x$) (see Methods for details). In Fig. 3, the time course (the first 14 s) of signal from each WM tract is displayed in the center of each panel with the time courses of the GM clusters that it connects vertically aligned at the top and bottom. The red line is the average time course across subjects. As expected, the time courses of the seven GM clusters were similar to one another, with a single peak arising at 6.14 ± 0.27 s. In contrast, the WM tracts demonstrated delayed responses, as well as a higher variance in the time to peak (TTP), which ranged from 8.58 to 10.00 s. The averaged peak magnitudes of the signal intensities in GM clusters were approximately 5.3 times those of WM tracts. Very similar results were obtained by simply averaging the time courses in the epochs following each incongruent event (Supplementary Fig. 1). Supplementary Fig. 2 shows the individual

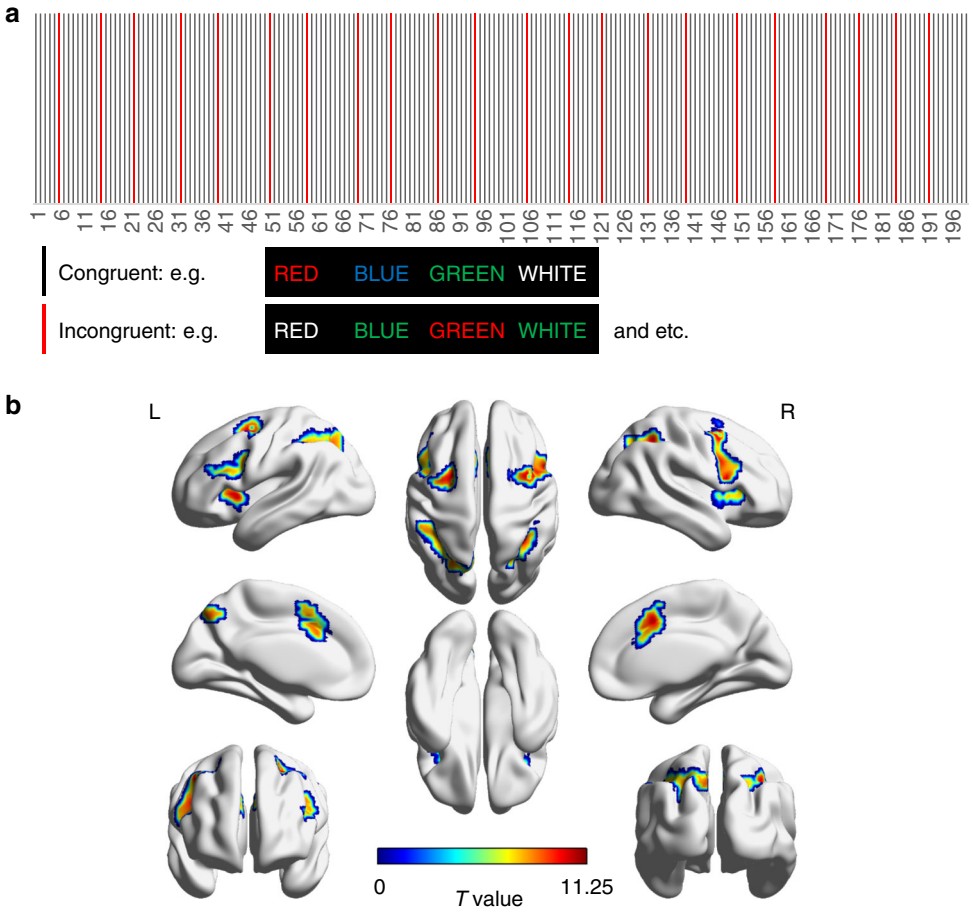

**Fig. 1** Experimental paradigms of the Stroop word-color task and the group activation map. **a** Experimental paradigms of the Stroop word-color task. Twenty-two incongruent words were shown randomly across each session with the interval between any two incongruent words no shorter than 14 s (=7 TRs). **b** Seven activated clusters (two sample *t*-test, FWE corrected) were symmetrically distributed in bilateral hemispheres, including the anterior cingulate gyrus, dorsomedial frontal cortex (dmPFC), bilateral middle frontal gyrus, bilateral inferior frontal gyrus, bilateral superior parietal gyrus, bilateral inferior parietal gyrus, bilateral precuneus, bilateral insular cortex and right angular gyrus

| Table 1 The names, coordinates and the overlapped structures of activated GM clusters | | |
|---|---|---|
| **Cluster name** | **Overlapped structures** | ***x, y, z* (mm)** |
| ACC | Anterior cingulate gyrus, dorsomedial-frontal cortex | −3 12 54 |
| Fron_L* | Middle frontal gyrus, inferior frontal gyrus left | −24 3 54−45 6 33 |
| Fron_R | Middle frontal gyrus, inferior frontal gyrus right | 36 6 51 |
| Insular_L | Insular left | −33 12 0 |
| Insular_R | Insular right | 27 21 6 |
| Par_L | Superior parietal gyrus, inferior parietal gyrus, precuneus left | −21 −63 42 |
| Par_R | Superior parietal gyrus, inferior parietal gyrus, precuneus, angular right | 36 −48 48 |
| *: denotes the area combined by two clusters that both overlapped with left middle/inferior frontal areas | | |

time courses of the eleven WM tracts in eleven selected typical subjects.

For comparison, event-unrelated time courses were obtained by the same process that yielded the event-related time courses after aligning the beginning of incongruent events to randomly selected time points (see Methods for definition). The event-unrelated time courses in WM tracts and their connecting GM clusters are displayed in Supplementary Fig. 3. All time courses had magnitudes that were not significant from zero and exhibited small variations with time that were unrelated to the timings of the incongruent stimuli.

**Statistical differences in HRF parameters between WM and GM.** For each WM tract, the TTP, magnitude, and area under

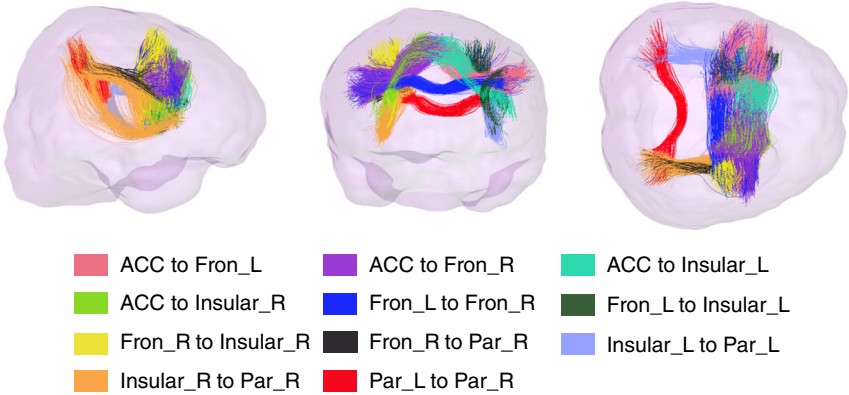

**Fig. 2** Eleven highly reproducible tracts. These tracts are displayed in different colors in a single subject's native coordinates

**Fig. 3** Average time courses in WM tracts and their connecting GM clusters. The time course of a WM tract was displayed in the center of each panel with the time courses of its connecting GM clusters vertically aligned at the top and bottom. The x-axis is the time (seconds), y-axis is the signal intensity (a.u.). Time = 0 is aligned to the onset of each incongruent stimulus. IQR = Interquartile Range. Source data are provided as a Source Data file

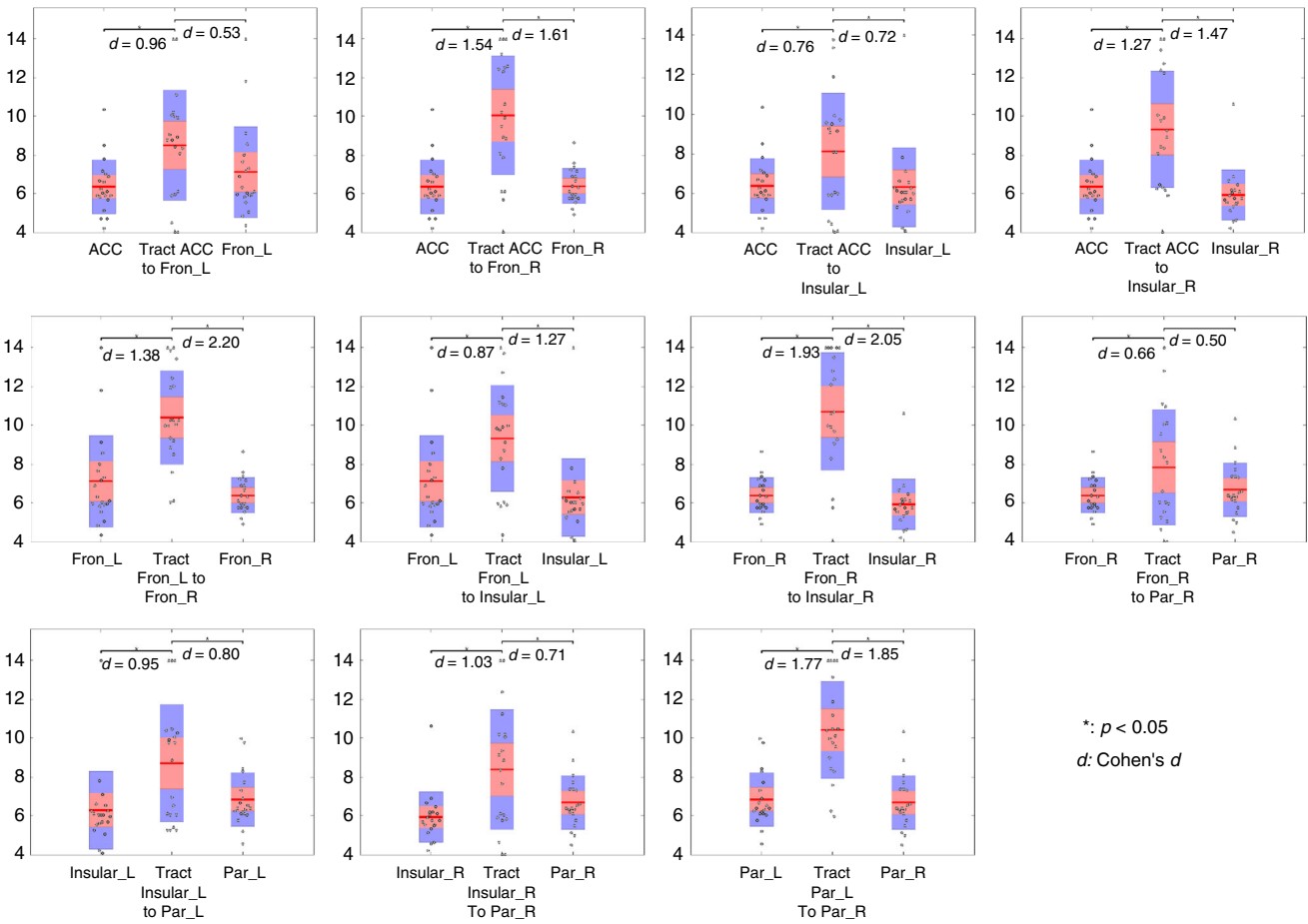

**Fig. 4** Average TTPs of WM tracts and their connecting GM clusters. Average TPP of one WM tract was displayed in the center of each panel with the average TTP of its connecting GM clusters on either side. The y-axis is the TTP (seconds). Points are laid over a 1.96 SEM (95% confidence interval) in red, a 1 SD in blue and a center line that denotes the median. 9 of 11 tracts showed significantly longer time ($P < 0.05$, $t$-test) to reach the peak in comparison with their connecting GM clusters. Source data are provided as a Source Data file

curve (AUC) were compared with its connecting GM clusters using two-sided $t$-tests (confidence intervals 95%) across the population. In Fig. 4, the average TTP of one WM tract is displayed in the center of each panel with the average TTP of its connecting GM clusters on either side. Points are laid over a 1.96 SEM (95% confidence interval) in red, a 1 SD in blue and a center line that denotes the median. It can be observed that 9 of 11 tracts required a significantly longer time ($P < 0.05$, $t$-test) to reach peak response in comparison with their connecting GM clusters. No significant differences in TTP could be found between the Fron_L and its connecting tract, which propagated into ACC, as well as between the Par_R and its connecting tract, which propagated into Fron_R. The peak magnitudes and AUCs of the time courses are displayed in Fig. 5 and Supplementary Fig. 4 respectively. Not surprisingly, a significantly lower magnitude and AUC were observed in all WM tracts compared with their connecting GM clusters ($P < 0.05$, $t$-test).

**Modeling the WM HRF with a modified double gamma function**. As a data reduction technique to quantify comparisons, we used a modified gamma model (see Methods for definition) to fit the average time courses of selected WM tracts that were delayed relative to their connecting GM clusters. The fitted parameters are displayed in Fig. 6. These curves, most of which showed pronounced apparent initial dips, were well fitted by double gamma function after the introduction of time delays into the first term. The estimated parameters of the double gamma

function and corresponding measurements of the fitted curve are listed in Supplementary Table 2. The parameters varied by tract, but the variance was limited to a small range among the group of tracts which shared similar curve shapes, e.g., (1) ACC to Fron_R, ACC to Insular_R, (2) ACC to Insurlar_L and Fron_L to Insular_L, (3) Fron_L to Fron_R, Insular_R to Par_R, and Par_L to Par_R. The measurements of the curves showed similar patterns to the estimated parameters but had smaller variance across all tracts in TTPs ($9.39 \pm 0.42$ s) and magnitudes ($0.87 \pm 0.15$ (a.u.)).

**Characterization of the HRFs in different depths of WM**. To characterize variations of HRFs with location within WM, the WM tracts were each subdivided into three sections, corresponding to superficial, middle and deep WM. Specifically, we created three WM masks by successively eroding the original WM outline three times (Supplementary Fig. 5). Next, the subsections of a WM tract were obtained by multiplying the tract and the masks. Note that the tracts analyzed in this study were not multiplied by the 95% WM mask to maintain the superficial WM voxels (see Methods Information for details). Two commissural tracts (Fron_L to Fron_R and Par_L to Par_R), which had larger numbers of deep WM voxels than other cortico-cortical tracts, were selected for this analysis. The HRFs with respect to three levels of WM tracts are shown in Fig. 7. It can be appreciated that the HRF in superficial WM voxels showed a shape more similar to that of GM but with lower magnitudes, and there was a trend towards a decreased magnitude, increased TTP and more

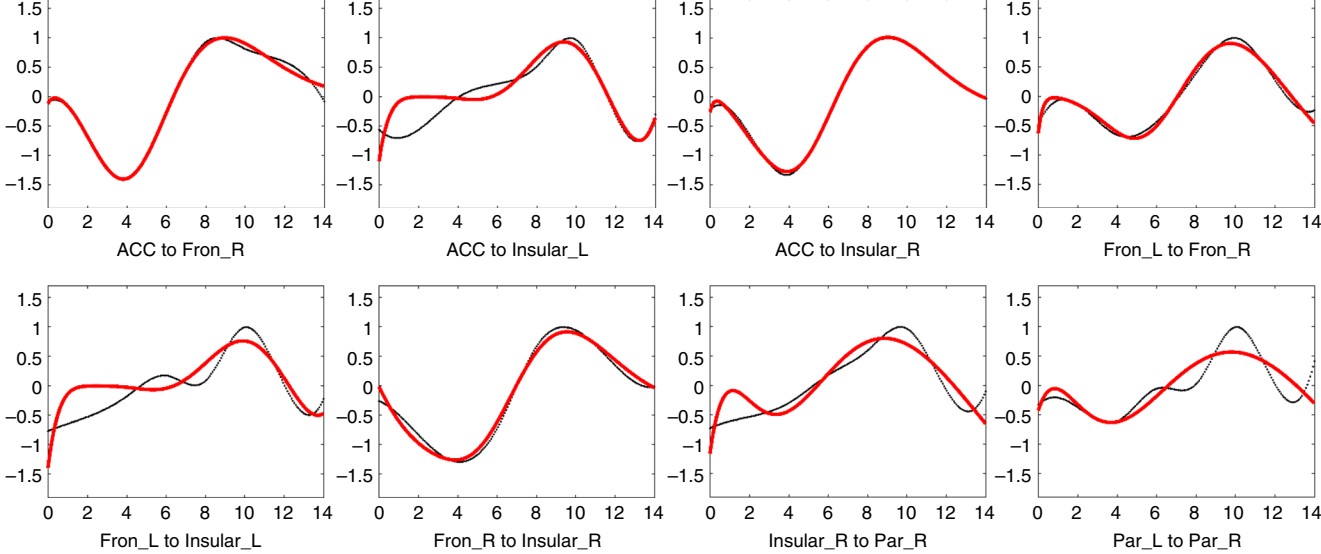

**Fig. 5** The average magnitudes of WM tracts and their connecting GM clusters. The average magnitude of one WM tract was displayed in the center of each panel with the average magnitudes of its connecting GM clusters on either side. The y-axis is the peak magnitude of the signal intensity (a.u.). Points are laid over a 1.96 SEM (95% confidence interval) in red, a 1 SD in blue and a center line that denotes the median. All WM tracts showed significantly decreased magnitudes ($P < 0.05$, t-test) in comparison with their connecting GM clusters. Source data are provided as a Source Data file

**Fig. 6** Fitting of the average time course of eight selected tracts. The x-axis is the time (seconds), y-axis is the signal intensity (a.u.). The black curves represent the average time courses. The red curves represent the fitting curves. Source data are provided as a Source Data file

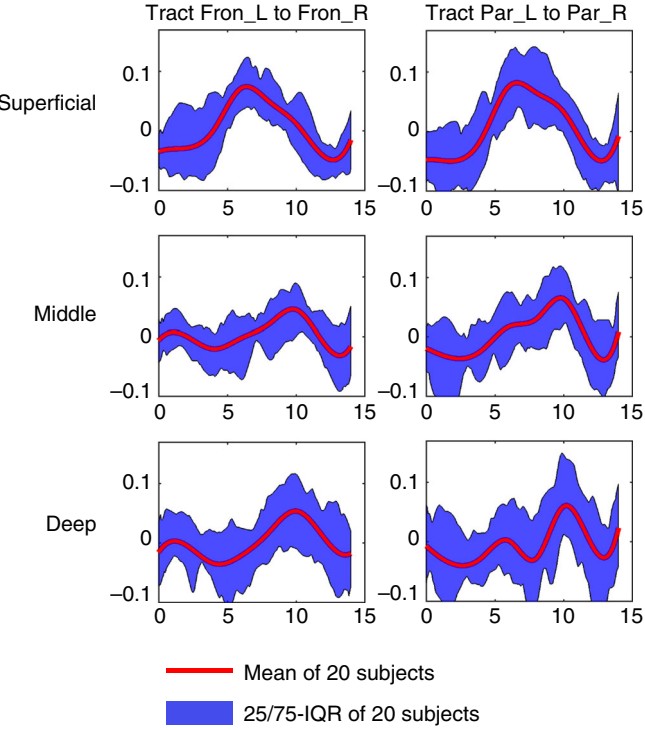

**Fig. 7** Average time courses in different depths of the WM tract. The *x*-axis is the time (seconds), *y*-axis is the signal intensity (a.u.). Time = 0 is aligned to the onset of each incongruent stimulus. IQR = Interquartile Range. Source data are provided as a Source Data file

prominent initial dip from the superficial to deeper sections of the WM tracts. The average magnitudes of these two HRFs are 0.08, 0.06, 0.06 (a.u.) from superficial, middle to deep WM, while the average TTPs are 6.51 s, 9.78 s, 10.00 s.

**Detection of WM activation using WM-specific regressor**. To test the relevance of the derived HRFs, we applied them to detect WM activations in selected WM bundles. Figure 8 displays an example of detected activations of subject-averaged BOLD data using both the canonical HRF model embedded in SPM (first row and second row) and our customized regressor reconstructed by convolving the incongruent stimuli time courses with the fitted HRF for the tract Fron_L to Fron_R (third row and fourth row). The WM activations (fourth row) are distributed in the anterior part of the body of the corpus callosum and anterior coronal radiation and are consistent with the known presence of fiber pathways that connect the activated frontal GM areas in the two hemispheres. On the contrary, these activations did not show up in the activation map based on the canonical HRF even with a loose threshold ($P < 0.02$, two sample *t*-test, uncorrected). When tightening the threshold ($P < 0.05$, two sample *t*-test, FWE corrected), the activation in GM tends to converge on the cluster center while the activation in WM is no longer detectable.

## Discussion
In this study, we characterized the HRFs of specific WM tracts that are connected with GM clusters in which neural activation was evoked by an event-related Stroop color-word task. We observed clear task-specific HRFs with reduced magnitudes and

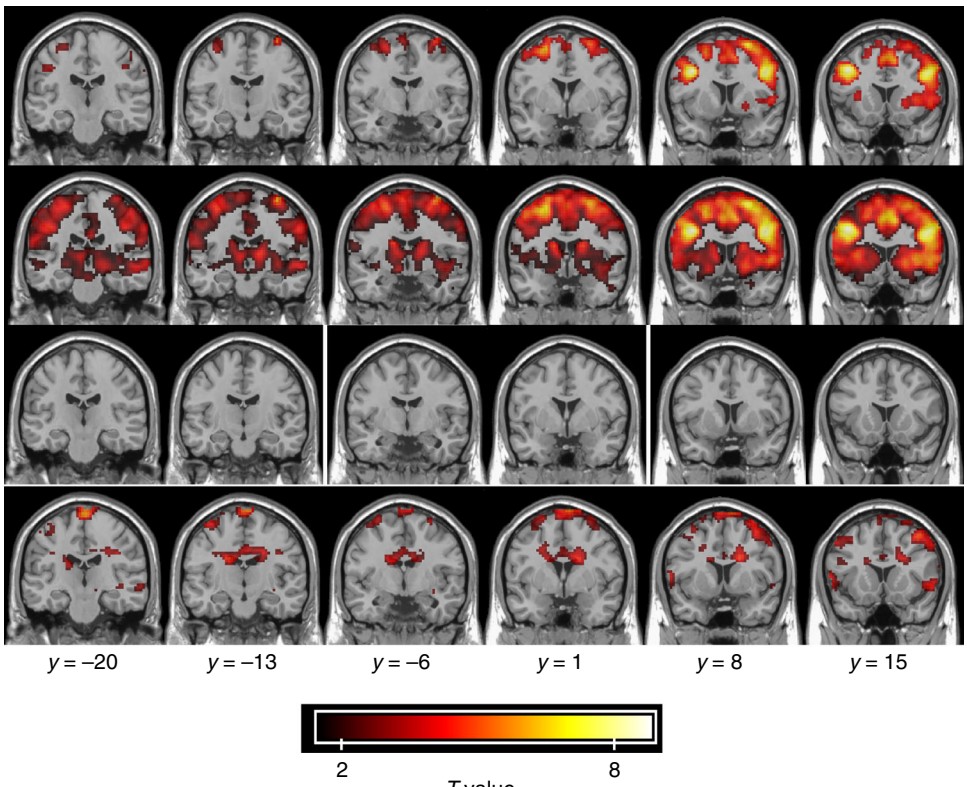

**Fig. 8** GM and WM activations detected using GM and WM HRFs. The first and second rows show the activation map based on canonical HRF model embedded in SPM (first row, $P < 0.05$, two sample *t*-test, FWE corrected; second row, $P < 0.02$, two sample *t*-test, uncorrected). The third row and fourth row show the activation map based on a custom regressor reconstructed by convolving incongruent stimuli with the fitted HRF for the tract Fron_L to Fron_R (third row, $P < 0.05$, *t*-test, FWE corrected; fourth row, $P < 0.02$, *t*-test, uncorrected)

increased TTPs in WM tracts compared with their connecting GM clusters, and the magnitude and TTP were related to the depth of the tracts[22]. Unlike the GM clusters, a significant portion of the WM tracts exhibited prominent negative initial dips such that their HRFs were manifested as biphasic profiles. The HRFs varied between tracts but were well fitted using the difference of two gamma functions.

The activated GM clusters detected in this study were consistent with previous findings of activations in the Stroop task distributed bilaterally throughout the brain, including the ACC, dmPFC, middle/inferior prefrontal areas, and insular and parietal areas[21,23–26]. The activated voxels in the ACC were concentrated in the dorsal portion, which has been considered to be involved in the monitoring of conflict between reading and the task of correctly naming the color[27]. Activation of the dmPFC was detected when subjects made choices that were inconsistent with their preferred strategy[28]. Moreover, the same investigators[29] observed a positive correlation between the dmPFC and insular cortex during compensatory choices. Additionally, both the prefrontal and parietal cortices were shown to be typically associated with executive function and decision making under risk and uncertainty[30,31]. Taken together, these findings indicate that the activated GM clusters were involved in several cognitive processes, including conflict monitoring, error correction, and suppression of automatic responses, which are consistent with the behavioral aspects of the Stroop color task.

In the current study, we used DTI tractography to investigate which of possible WM connections between every pair of activated clusters showed a synchronous BOLD change with the task. The reproducibility of the tractography for 11 tracts from all 20 subjects was very high and allowed measurement of the HRF to be reliably detected. The time courses of BOLD signals in those WM tracts clearly showed a task-specific, synchronized, transient response with a significantly decreased magnitude and AUC but increased TTP in comparison with connecting GM clusters. By design, these WM responses cannot be attributed to potential confounding effects from GM areas because we used three constraints to minimize any such effects. First, the WM analysis was based on unsmoothed fMRI images, which preserved the boundaries between the WM and GM. Second, using the DTI measurement, the WM voxels were selected at threshold FA > 0.3 to exclude the GM, which has much lower diffusion anisotropy with FA ranging from 0.15 to 0.24[32,33]. Third, the probabilistic WM mask, obtained by segmenting the T1 images of each subject, was used to filter out GM voxels, as well as superficial WM voxels, at a tight threshold (>95%).

A common concern is that the observed WM activations may be spurious because of residual effects from oxygenation changes in GM vasculature. Anatomically, WM receives blood supply almost entirely from the medullary artery[34], which originates directly from the pial artery and does not give off branches to cortical areas[35]. Although the arteries supplying WM pass through cortical GM, it seems highly unlikely that BOLD signal changes produced by activation of the GM could have much impact on "downstream" WM because the GM receives blood through a separate, dedicated arterial system rather than through the arteries that penetrate the cortex to supply WM. In fact, it has been experimentally demonstrated in a rat model that increases in cerebral blood flow in response to neural activity are tightly restricted to the site of activation, with sub-millimeter precision[36,37]. With respect to the venous system, a large activated GM cluster may generate oxygenation changes along draining veins up to several millimeter beyond the edge of the activated cluster[38]. However, unlike cortical areas, whose venous drainage collects into the pial veins located at the surface of the brain, deep WM tracts are drained via the sub-ependymal veins,

which are close to the lateral ventricles[39]. As such, there are no vascular interactions between the two different tissue types, and the blood flow out of activated GM regions is unable to reach deep WM to modulate the signals therein (exceptions exist for cases of developmental venous anomalies but these have an incidence only up to 2.6%[40]). Another potential concern is that the signal fluctuations we observed in WM might not be purely BOLD in origin because factors other than the level of oxygenation could modulate local magnetic susceptibility as well. For example, the progressively increased time delay of signal changes in some WM tracts could be partly attributable to the influence of susceptibility changes in parallel vessels, in which half of the WM blood resides[41]. However, such potential confounding factors should tend to shorten transverse relaxation times and thus would decrease the magnitude of MRI signals, so it is unlikely that such factors account for the positive peaks of the BOLD signals we observed, though in principle they could contribute to the initial negative dips.

One of our most important findings is that WM HRFs show reduced magnitudes and delayed responses compared with GM. Similar differences have been observed in cerebrovascular reactivity (CVR) studies after hypercapnic challenges[14,42–44]. The CVR depends only on vasoactive responses, rather than neural activity, but these findings are consistent with a BOLD effect in WM corresponding to lower vascular densities and longer distances between feeding arterioles and areas of increased oxygen use. Another important finding is that 5 of 11 tracts showed pronounced apparent initial dips in their HRFs. By analogy with the elusive initial dips reported in GM[45–47], these may represent early focal increases in oxygen usage that are relatively larger and which induce negative BOLD signals for longer than in GM, before the arterial supply increases flow sufficiently to meet the tissue demand. Due to this complex shape, we fit the WM HRF with a mathematical expression that incorporates the subtraction of two gamma functions. To account for the delayed flow increases in WM tracts, we added an extra parameter, $t$, to indicate the time lag of the curve. Our data show that the HRFs, with or without initial dips, were well-fitted. These analyses result in fitting parameters that are distinctly different from those that characterize the canonical HRF of GM, thus calling for significant changes to the standard methods used for more effective detection of neural activities in WM. The quantitative fitting is useful as a data reduction technique but may also be used as the basis for physiological models that attempt to interpret the coupling between cerebral blood flow (CBF) and cerebral metabolic rate of oxygen (CMRO2) in WM. To better capture these physiological underpinnings, however, the WM HRFs likely need to be characterized using more complex expressions in future work, to understand the BOLD signal as a combination of multiple physiological processes, as, for example, are included in a model based on prompt arterial dilation and the general hypothesis of proximal integration[48–50]. In superficial WM voxels, the HRF showed a shape more similar to that of GM possibly because the GM and superficial WM share the same vascular system. The duration of the initial dip progressively increased, both in magnitude and width, from superficial to deep WM. A plausible explanation is that in deep WM it takes longer to compensate for an increase in $O_2$ consumption because of the nature and distance of the supplying vessels[15] so that deoxyhemoglobin continues to accumulate until oxygenated blood arrives.

We have demonstrated that the activation of deep WM was not detectable when using the canonical HRF for GM even with loose constraints, but could be detected with our customized regressor derived from a tract-specific HRF. However, it is also worth emphasizing that WM activity will often not show up if the same

criteria (threshold and cluster size) are applied as are used for GM BOLD effects because they are substantially weaker.

We recognize that, in this study, we acquired images with relatively high SNR and GM-WM contrast and low image distortion, but with a spatial resolution which could have introduced some partial volume effects. With the recent rapid evolution of Simultaneous multi-slice sequences and maturation of reliable imaging protocols, more favorable trade-offs between contrast/ signal-to-noise ratio and spatial/temporal resolutions may be achieved.

In conclusion, we investigated HRFs of those WM pathways which are structurally connected with regions of GM activity engaged in the cognitive response in the event-related Stroop color task. The WM HRFs exhibited decreased magnitudes, increased times to peak and strong task specificity. By fitting the data to modified gamma functions, a set of parameters was derived to characterize the HRF curve which may be used to further explore the vascular mechanism underlying BOLD signals in WM tracts. Taken together, these finding demonstrated the detectability of neural activities in WM and, more importantly, reveal the nature of WM HRFs and emphasize the need to use alternate data analyses in seeking WM activity in conventional fMRI.

## Methods

**Subjects**. The present study was approved by the Vanderbilt University Institutional Review Board. Written informed consent was obtained from each subject. Twenty healthy and right-handed individuals (10 M/10 F; age, $29.1 \pm 8.8$ yrs) with no history of neurological or psychiatric disorders were recruited.

**Experimental design**. The Stroop test relies on the observation that color naming of a printed word can be slowed by the attendant presence of a color word[21]. For example, naming the color of a word "BLUE" displayed in red color (incongruent word) is often slower than naming the word "BLUE" displayed in blue color (congruent word). This behavior can be explained by the conflict between the need to inhibit the automatic reading of a word in favor of naming the color. In the current study, the color words were generated through the combination of four words ("RED", "GREEN", "BLUE", and "WHITE") and four colors (red, green, blue, and white). Two hundred color words were successively presented throughout each session (400 s) with each word displayed against a black background for 1700 ms. Each word was followed by an interval of 300 ms, during which a gaze-fixation (a white crosshair) was displayed at the center of the screen. As shown in Fig. 1a, twenty-two incongruent words were shown randomly across each session with the interval between any two incongruent words no shorter than 14 s (=7 TRs). Subjects were instructed to name the color of every word silently as rapidly as possible and provide feedback by clicking a button in response to the color: index finger = red, middle finger = green, ring finger = blue, and pinky finger = white. E-prime (Psychology Software Tools, Inc) was used for presenting the visual stimuli and receiving feedback from the subjects. All subjects were asked to view an instructional video before the scanning to fully understand the Stroop test. The task began with a practice session that consisted of 20 stimuli, during which the subjects' response times and accuracy rates were displayed on the screen to ensure compliance.

**MRI acquisition**. A 3T Philips Achieva scanner (Best, Netherlands) installed at Vanderbilt University Institute of Imaging Science with a 32-channel head array coil, was used in this study. Each subject was scanned in a supine, head-first position with restricting pads placed within the head coil to ensure stability. The fMRI images were acquired from these subjects with TR = 2 s, TE = 35 ms, SENSE factor = 2, matrix size = 80 × 80, FOV = 240 × 240 mm$^2$, 34 slices of 4 mm thickness with a 0.5 mm gap, and 200 dynamics. Simultaneous multi-slice (SMS) sequences, though yielding higher spatial resolution with comparable acquisition times, were not employed in the current study as our pilot data indicated that SMS tended to reduce SNR in BOLD signals as well as gray-white matter contrast, in addition to introducing greater geometric distortions. The color words were visually projected to a screen mounted in the back of the scanner bore and could be viewed by subjects through a mirror mounted on the head coil. A right-hand button unit was used to collect real-time feedback from each subject.

To reconstruct white matter tracts, diffusion-weighted MR images were acquired using a multi-shot, echo-planar imaging (EPI) sequence with $b = 1000$ s per mm$^2$, 32 diffusion-sensitizing directions, TR = 4.5 s, TE = 84 ms, matrix size = 112 × 112 × 68, and voxel size = 2 × 2 × 2 mm$^3$. As anatomical references, high-resolution T1-weighted images were acquired using a three-dimension (3D)

magnetization-prepared rapid gradient-echo (MP-RAGE) sequence at voxel size of 1 × 1 × 1 mm$^3$.

**fMRI analysis**. fMRI images were preprocessed using the statistical parametric mapping (SPM12) software (www.fil.ion.ucl.ac.uk/spm/software). First, the fMRI images were corrected for within-scan acquisition time differences and inter-scan head motions. Second, for each subject, GM, WM, and cerebrospinal fluid (CSF) masks were obtained by segmenting the T1 structural image. Next, these masks, as well as T1 images, were co-registered to the fMRI coordinates for the same subject. All resulting images, including the fMRI, T1, and GM/WM/CSF masks, were then spatially normalized to a standard space (Montreal Neurological Institute (MNI)) coordinates, at a voxel size of 3 × 3 × 3 mm$^3$. The voxel-wise timecourses extracted from the normalized fMRI images were subsequently temporally filtered by a high-pass filter (128 s) and normalized to zero mean and unit variance. Finally, the resulting images were spatially smoothed with a Gaussian kernel (full-width at half maximum 6 × 6 × 6 mm$^3$). Note that un-smoothed images were preserved for the time course analysis to minimize mutual influences between GM and WM BOLD signals. Confounding effects of physiological fluctuations, such as cardiac pulsations and respiration-induced modulations, on fMRI time-series were removed using a CompCor approach[51]. Specifically, nuisance variables were derived from the anatomical noise ROI, identified within cerebrospinal fluid (CSF), and were subsequently regressed out from the BOLD time series. Following preprocessing, the task conditions (incongruent events and congruent events) were convolved with a canonical HRF in the context of a general linear model integrated within SPM. On a single subject level, conditions were contrasted against each other to create a parametric image that reflected the percent signal changes of incongruent events compared with that of the congruent events. On the group level, a one-sample $t$-test was applied to the parametric images across all subjects to create a map of the additional brain activation produced by the incongruent responses for the whole population. Activated voxel clusters were reported at a threshold $P < 0.05$ (cluster level, two sample $t$-test, family-wise error rate (FWE) corrected) with the cluster size larger than 50 voxels.

**DTI analysis**. The raw diffusion-weighted images (DWIs) were first co-registered to B$_0$ images and corrected for eddy current and subject motion with affine transformations using Automated Image Registration (AIR)[52]. The 3 × 3 diffusion tensor were calculated for each pixel with multivariate linear fitting. For each subject, the B$_0$ image was co-registered to the T1 images and subsequently normalized to MNI space using SPM. This procedure was reciprocal and provided forward (DTI space towards MNI) and backward (MNI towards DTI space) transformation matrices. To guide the tracts reconstruction, the activated clusters of each subject were transformed from MNI to the DTI space using the backward transformation matrix. Between each pair of the activated clusters, we explored the possible connections by performing fiber tractography using an in-house Matlab script based on a stochastic fiber tracking approach[53]. The tracts which were reproducible among all twenty subjects were preserved for further analysis. These tracts, in the form of coordinate sets, were converted to binary image files (1 assigned to voxels that contain the tracts and with fractional anisotropy FA > 0.3; 0 assigned to the remaining voxels) and were later transformed to the MNI space using the forward transformation matrices. In MNI space, the binary tracts of each individual were then masked with the WM mask at a threshold of 95% to exclude the influences from adjacent GM, as well as superficial WM voxels.

**Time course analysis**. The time courses within the WM tracts and activated GM clusters were extracted from the un-smoothed fMRI images and sorted to align with the timings of the incongruent stimuli. Each subject-specific hemodynamic response $h$ following the incongruent events could be obtained by using a deconvolution approach based on the fMRI time course $y$ and the know stimuli vector $x$. Several measurements, including the magnitude, time-to-peak (TTP) and area under the curve (AUC), were calculated to describe the response curves. The mean measurements were compared between GM and WM across all subjects using a two-sample $t$-test. To confirm the task-specificity of the resulting time courses, time courses not related to events were produced as comparison. Specifically, 22 time points were randomly sampled (with repeats) throughout the session. A bootstrap estimator could be generated by applying the deconvolution approach to the fMRI time course based on the randomly sampled stimuli. The event-unrelated time course was obtained by repeating this process 1000 times and then averaging the estimators.

In order to parameterize the HRFs, the time courses were fit to a modified double gamma function. Each time course was first normalized by dividing by the maximal value over the time window. After that step, the curves were fit to equation (1), the difference of two gamma functions. Each gamma function can be parameterized in terms of $c$, the amplitude, $a$, a shape parameter, $b$, a scale parameter, and $t$, a time lag.

$$f(x) = c1 * \frac{(x-t1)^{a1-1}}{b1^{a1} * (a1-1)!} * e^{\frac{-(x-t1)}{b1}} - c2 * \frac{(x-t2)^{a2-1}}{b2^{a2} * (a2-1)!} * e^{\frac{-(x-t2)}{b2}} \quad (1)$$

Finally, to measure variations in the neurovascular coupling at different depths of WM, the tracts were subdivided into three sections, from which the HRFs could be obtained separately from superficial, middle and deep WM. To subdivide the tracts,

we successively eroded the WM mask three times to create specific masks (Level 1 to Level 3) consisting of voxels at different depths. Specifically, the Colin 27 Average Brain (CH2) WM template[54] (MNI space, 1 mm voxel resolution) was taken as the initial input of the erosion operations. A sphere-shaped element with radius of 4 mm was created to probe the input image and define the neighborhood used in the erosion of each pixel. The Level $i$ mask was obtained by subtracting the output image from the input image during the $i$th iteration of erosions. Finally all the high-resolution masks were resliced to 3 mm voxel resolution to match the coordinates of the normalized fMRI images. Next the subsections of a WM tract could be obtained by multiplying the tract and the masks. The average time courses with respect to each of the three sections of tracts were extracted and compared.

**Reporting summary**. A reporting summary for this Article is available as a Supplementary Information file.

## Code availability

The study was developed based on Matlab (MathWorks, Inc) using a combination of freely available tools that listed below and custom code. SPM12, https://www.fil.ion.ucl.ac.uk/spm/software/spm12/ REST V1.8, http://restfmri.net/forum/index.php DTIstudio V3.03 and Diffeomap V1.9, https://www.mristudio.org/.

## Data availability

The datasets that generated Figs. 3–7, Supplementary Figs. 1–4 are provided as a Source Data file and are also available in the figshare repository, https://doi.org/10.6084/m9.figshare.7451015.v2.

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

## Acknowledgements

This work was supported by NIH grant R01 NS093669 (J.C.G).

## Author contributions

J.C.G., Z.D. and A.W.A. supervised the project; J.C.G., Z.D. and M.L. conceived the experiments; M.L., A.T.N. and D.Z. conducted the experiments; M.L. analyzed the data; M.L., D.Z. and J.C.G. wrote the manuscript. All authors discussed the results.

## Additional information

**Competing interests:** The authors declare no competing interests.

