## [Peer Review File · Nature Communications]

Reviewers' comments:

Reviewer #1 (Remarks to the Author):

Li and colleagues characterize functionally evoked hemodynamic changes observed in white-matter tracts. Activation was evoked by color-name incongruence during a Stroop color-word identification task. Strong activity was observed in several gray-matter brain regions, such as medial frontal gyrus and anterior cingulate. Additionally, weaker but significant responses were observed in white-matter voxels corresponding to various tracts linking these gray-matter regions. These responses exhibit a delayed time-to-peak as compared to gray-matter HRFs. Authors suggest that these hemodynamic changes are directly linked to functional activity within the white matter by poorly understood neurovascular coupling mechanisms. These results are fascinating because white-matter neurovascular coupling is poorly understood and appreciated. The ability to associate specific white-matter activity with experimental manipulations is a fascinating manifestation of neurovascular coupling, and a potentially powerful new tool for neuroscience research in general. The manuscript is clearly and compactly written at an appropriate level for general audiences.

There are, however, a few issues that need to be addressed to make the results sufficiently compelling. Foremost, the experimental design is a randomized slow event-related design with minimum ISI of 14 seconds. The analysis of the time series is direct averaging. For gray-matter HRFs, this approach is acceptable, because their temporal form is well known to be fairly compact, so that late-time aliasing will be relatively small. On the other hand, the white-matter HRFs are essentially unknown, so such assumptions are not justified. In fact, the current analysis suggests that white-matter HRFs rise to a peak much later than in the corresponding gray matter, and it is unclear if they exhibit additional late-time behavior that extends beyond the 14-second window. The authors could improve matters by applying a deconvolution approach to the data, making the usual assumption of shift-invariant linearity. Such an analysis would bolster that notion that the current analysis largely recovers the relevant dynamics of the white-matter HRFs. A second matter concerns the general weakness and variability of the white-matter HRFs. Although the mean time-course signals across subjects appear significant, it would be comforting to see some examples of individual subjects with significant responses, and perhaps a table summarizing responses from all individuals. Finally, the attempt to relate the measured results to the "balloon model" are flawed in several ways. To begin with, the use of a difference-of-gamma-variate functions as a model has no real relationship to any theory; this is strictly a heuristic fitting procedure that has become popular. The attempt to somehow use the success of this fit as a measure of its theoretical credibility is silly. Moreover, the balloon model is now obsolete. Study after study in animal models using various form of imaging have failed to demonstrate significant venous vasodilation. Newer models for the BOLD HRF in gray matter have attempted to resolve this issue based on the clear evidence for active arterial vasodilation and the general hypothesis of proximal integration as a mechanism for neurovascular coupling. These modern theories need to be incorporated into the Discussion in some form.

A more minor issue concerns the general choice of scanning protocol. Why choose such low spatio-temporal resolution for the fMRI protocol? Modern scanners generally can obtain 2-mm cubic voxels with 1.5-s TR using only modest SMS acceleration. The choice of the rather archaic fMRI protocol at the generally state-of-the-art Vanderbilt imaging center seems hard to justify. Were there worries that SMS would introduce artifacts in the white-matter responses?

Reviewer #2 (Remarks to the Author):

Dear authors

Thanks a lot for submitting this interesting paper.

Investigating the hemodynamic response function in white matter is definitely an interesting and challenging topic, and your approach of looking at traits connecting activated regions is original and promising.

This said, there are some caveats and confounders which could be addressed more clearly, to properly interpret your results.

In your paper there is not a clear interpretation of the functional role of the HRF in the white matter tracts connecting two activated areas. You talk about signal travelling, but from where to where? In both directions? Unidirectionally?

You say that you can exclude that the stimulus-locked hemodynamic activity in white matter is due to the flux of blood to activated areas of grey matter. You bring some arguments towards this interpretation, but on the other hand we cannot ignore that the vascular system is unique and closed, and that blood arrives to different regions in the brain with some time delays.

Proper physiological control and correction would be in order (breath holding, measurement of cardiac pulse and respiration).

Time-locked activity is been found basically in all the brain if you look for it

(<http://www.pnas.org/content/109/14/5487>,

<https://academic.oup.com/cercor/article/25/12/4667/309934>). This makes less evident your

choice of looking at specific regions and tracts, unless you don't validate somehow that the HRF in the tracts reflects the communication between the connected areas. This communication, if it exists, can be neuronal or physiological, in particular if we know already that we find HRF in white matter, and that your main contribution would be the causal role of the GM activation in the shape and existence of the HRF one. Your evidence is purely anecdotal, there is no validation of this hypothesis, nor a counterhypothesis. You should for example look at HRF everywhere in the brain, and see if the shape in WM is modulated by the activation of connected areas, possibly with different paradigms.

It has been recently confirmed that brain areas involved in task enlarge

<https://www.biorxiv.org/content/early/2018/09/05/408658> . If this would be the case in your

data, it could be that the grey matter would expand where the white matter is supposed to be in your mask.

Some results and open maps exist from a data driven approach to retrieve a proxy of the hemodynamic response function at rest, including extended to white matter

<https://neurovault.org/collections/3584/>

<https://doi.org/10.6084/m9.figshare.7139702.v2>

You could possibly compare your results with those, to obtain a sort of a baseline.

Why using the gamma/balloon models and not a Finite Impulse Response, or an approach based on physiological Gaussian priors (<https://www.biorxiv.org/content/early/2017/11/14/179838>)?

On page 7 you write: "Such connectivity patterns may provide the functional architecture for signals to be transmitted from one node to another among these activated clusters". This is a strange and recursive definition: there are coactivated areas, which come up with ICA or seed-based correlation, and we call them Functional Connectivity, as opposed to the Structural one. At these scales we don't need FC to infer SC, and there are many factors, including vascular and autonomic physiology (and movement) which contribute to the formation of FC patterns.

Figures: I appreciate that you show all the data points in the box plots. On the other hand p-values are an uniform distribution, no need of discretizing them with asterisks indicating different thresholds. Also, p-values should be accompanied with an effect size. If you have an hypothesis of the link between the activations in GM regions and connecting tracts, you should include it in the figure (the simplest model (i.e. no model) would imply just to link all the triplets of points for the same subject).

To summarize, it's likely that what you measure is actually hemodynamic response functions in white matter (the BOLD comes from vessels, mainly microvessels, see this recent study

<https://www.ncbi.nlm.nih.gov/pubmed/29398359>), and there are vessels surrounding white matter (even though for this type of study I would have expected a more rigorous account of the role of respiration and cardiac phase). But you cannot exclude that what you measure comes (all or in part) from vessels in the grey matter.

What is not clear is that the white matter HRF that you find is a signature of a signal travelling

between two activated areas, neither of any other sort of communication between them, apart from the mere existence of the white matter tracts.

Reviewer #3 (Remarks to the Author):

Li et al. present an excellent study investigating white matter (WM) activation in functional MRI. I know the area well and was impressed with the quality of the experimental work and the drafted manuscript.

Major suggestion: Recommend the authors add a figure show WM activation using the novel HRF model parameters to significantly improve impact.

Minor suggestions: One never wants to request additional citations related to our own work, but in this case I suggest that the authors consider adding these references:

1) In the initial WM fMRI reference (Lines 54-55) - Gawryluk et al., 2014 Frontiers Review paper - as this provides source material that will help substantiate WM HRF physiology interpretations and supporting literature.

2) Fraser et al. (2012) BMC Neuroscience and Yarkoni et al (2009) PloS One - as both of these specifically focus on WM HRF characteristics (in addition to Courtemanche et al.).

Also, lines 166 and 167 are speculative and best moved to Discussion section rather than Results section.

Reviewer #1 (Remarks to the Author):

Li and colleagues characterize functionally evoked hemodynamic changes observed in white-matter tracts. Activation was evoked by color-name incongruence during a Stroop color-word identification task. Strong activity was observed in several gray-matter brain regions, such as medial frontal gyrus and anterior cingulate. Additionally, weaker but significant responses were observed in white-matter voxels corresponding to various tracts linking these gray-matter regions. These responses exhibit a delayed time-to-peak as compared to gray-matter HRFs. Authors suggest that these hemodynamic changes are directly linked to functional activity within the white matter by poorly understood neurovascular coupling mechanisms. These results are fascinating because white-matter neurovascular coupling is poorly understood and appreciated. The ability to associate specific white-matter activity with experimental manipulations is a fascinating manifestation of neurovascular coupling, and a potentially powerful new tool for neuroscience research in general. The manuscript is clearly and compactly written at an appropriate level for general audiences. There are, however, a few issues that need to be addressed to make the results sufficiently compelling.

Response: We greatly appreciate your positive evaluation of this work. Our point-by-point response follows each question below.

1. Foremost, the experimental design is a randomized slow event-related design with minimum ISI of 14 seconds. The analysis of the time series is direct averaging. For gray-matter HRFs, this approach is acceptable, because their temporal form is well known to be fairly compact, so that late-time aliasing will be relatively small. On the other hand, the white-matter HRFs are essentially unknown, so such assumptions are not justified. In fact, the current analysis suggests that white-matter HRFs rise to a peak much later than in the corresponding gray matter, and it is unclear if they exhibit additional late-time behavior that extends beyond the 14-second window. The authors could improve matters by applying a deconvolution approach to the data, making the usual assumption of shift-invariant linearity. Such an analysis would bolster that notion that the current analysis largely recovers the relevant dynamics of the white-matter HRFs.

Response: Thank you very much for pointing this out. Per your suggestion, we employed the deconvolution approach to re-estimate the hemodynamic response function that predicts measured fMRI signals from a known event vector (stimuli). We observed that there are no noticeable differences between the time courses (first 14 seconds) obtained using the deconvolution approach and those from direct averaging. According to your suggestion to use this approach, we updated the relevant methods section and results including Fig. 3 – Fig. 7, Supplementary Fig. 2- Fig. 4 and Supplementary Table2. In addition, we also present our original epoch-averaging method results in the Supplementary section for comparison which avoids the assumption of linearity (Supplementary Fig. 1).

2. A second matter concerns the general weakness and variability of the white-matter HRFs. Although the mean time-course signals across subjects appear significant, it would be comforting to see some examples of individual subjects with significant responses, and perhaps a table summarizing responses from all individuals.

Response: We selected eleven typical subjects, and displayed in Fig. R1 their individual time courses of the eleven WM tracts that reproducibly connects the activated GM clusters. We added this figure to the supplementary information as well (Supplementary Fig. 2). Meanwhile, the variations in Fig. 4 and Fig. 5 encode the response characteristics, including TTPs and magnitudes, of all individuals.

Fig. R1. White matter time courses of eleven WM tracts in selected subjects

3. Finally, the attempt to relate the measured results to the “balloon model” are flawed in several ways. To begin with, the use of a difference-of-gamma-variate functions as a model has no real relationship to any theory; this is strictly a heuristic fitting procedure that has become popular. The attempt to somehow use the success of this fit as a measure of its theoretical credibility is silly. Moreover, the balloon model is now obsolete. Study after study in animal models using various form of imaging have failed to demonstrate significant venous vasodilation. Newer models for the BOLD HRF in gray matter have attempted to resolve this issue based on the clear evidence for active arterial vasodilation and the general hypothesis of proximal integration as a mechanism for neurovascular coupling. These modern theories need to be incorporated into the Discussion in some form.

Response: Thanks to the reviewer for this suggestion. Conventionally, the balloon model is proposed to explain the dynamics of microvascular response to neural activity, which is still seen in recent literatures (e.g., Özbay, et al [1301]). We recognize that many studies have demonstrated that CBV changes from neural activations originate primarily from the arterial rather than venous blood volume changes, which renders the balloon model interpretation obsolete. Therefore, we removed the relevant text regarding the balloon model whose mechanisms are not fully understood. However, we have retained the data fitting as the linear combination of two gamma functions in the manuscript, which provides an elegant mathematical framework for capturing the large initial dip and peak latencies of WM response and is a potentially useful data reuction method. Our purpose here is not to interpret the BOLD signal as a combination of multiple physiological processes, but to derive distinct sets of parameters which could be a supplement to canonical HRF. In future work, we hope to fit the WM HRF using

physiologically more realistic models, for example, the model based on arterial dilation and proximal integration. Relevant discussions have been added to the manuscript, copied below:

“Due to this complex shape, we fit the WM HRF with a mathematical expression that incorporates the subtraction of two gamma functions. To account for the delayed flow increases in WM tracts, we added an extra parameter, t , to indicate the time lag of the curve. Our data show that the HRFs, with or without initial dips, were well-fitted. These analyses result in fitting parameters that are distinctly different from those that characterize the canonical HRF of GM, thus calling for significant changes to the standard methods used for more effective detection of neural activities in WM. The quantitative fitting is useful as a data reduction technique but may also be used as the basis for physiological models that attempt to interpret the coupling between cerebral blood flow (CBF) and cerebral metabolic rate of oxygen (CMRO₂) in WM. To better capture these physiological underpinnings, however, the WM HRFs likely need to be characterized using more complex expressions in future work, to understand the BOLD signal as a combination of multiple physiological processes, as, for example, are included in a model based on prompt arterial dilation and the general hypothesis of proximal integration [1302].”

1301. Pinar S. Özbay, Catie Chang, Dante Picchioni, Hendrik Mandelkow, Thomas M. Moehlman, Miranda G. Chappel-Farley, Peter van Gelderen, Jacco A. de Zwart, Jeff H. Duyn, Contribution of systemic vascular effects to fMRI activity in white matter, *NeuroImage*, Volume 176, 2018, Pages 541-549.

1302. Kim, J. H., Khan, R., Thompson, J. K., & Ress, D. (2013). Model of the Transient Neurovascular Response Based on Prompt Arterial Dilation. *Journal of Cerebral Blood Flow & Metabolism*, 33(9), 1429–1439.

4 A more minor issue concerns the general choice of scanning protocol. Why choose such low spatio-temporal resolution for the fMRI protocol? Modern scanners generally can obtain 2-mm cubic voxels with 1.5-s TR using only modest SMS acceleration. The choice of the rather archaic fMRI protocol at the generally state-of-the-art Vanderbilt imaging center seems hard to justify. Were there worries that SMS would introduce artifacts in the white-matter responses?

Response: We are fully aware of recent advances in SMS technology. While our institute enthusiastically embraces new and advanced technologies, we are also quite conservative in employing them since it is not uncommon that they may bring harmful artifacts. In particular relevance to the current study, the SMS seems to reduce gray-white matter contrast and SNR in BOLD signals, in addition to introducing greater geometric distortions (Shown in Fig. R2 are examples of our data and images from the human connectome project with TR = 0.72 s, voxel size = $2 \times 2 \times 2$ mm³, Multi-slice acceleration = on). The contrast between gray and white matter is important to this study because inaccuracies in white matter boundary may subject our findings to partial volume effects from adjacent gray matter, one of the common concerns over our work. Also, SNR in white matter is very important because BOLD signals therein are relatively weak, which renders their detection rather difficult.

Fig. R2. BOLD images acquired in the current study (top row) and images obtained from human connectome project (bottom row)

We recognize that, with rapid evolution of SMS technology and maturation of reliable imaging protocols, optimal trade-offs between contrast/signal-to-noise ratio and spatial/temporal resolutions could be achieved. In the near future, we will systemically evaluate the performance of the SMS technology, and employ an optimized protocol in our functional studies.

Reviewer #2 (Remarks to the Author):

Dear authors

Thanks a lot for submitting this interesting paper.

Investigating the hemodynamic response function in white matter is definitely an interesting and challenging topic, and your approach of looking at traits connecting activated regions is original and promising. This said, there are some caveats and confounders which could be addressed more clearly, to properly interpret your results.

Response: We greatly appreciate your evaluation of this work. Our point-by-point response follows each question below.

1 In your paper there is not a clear interpretation of the functional role of the HRF in the white matter tracts connecting two activated areas. You talk about signal travelling, but from where to where? In both directions? Unidirectionally?

Response: Thank you very much for pointing this out. We apologize for the misleading statement about signal traveling between GM and WM. The reason we mentioned the role of WM as the signal carrier was to introduce the notion that neural activity exists in WM and may also produce detectable BOLD signals. Our recent publication [2101] demonstrated that specific WM voxels are robustly activated, and that regional distributions of those WM voxels are consistent with fiber pathways that connect the activated GM areas. Therefore, the purpose of tract-based analysis of BOLD signals is to group WM voxels that are presumably activated by the same events as those activating the GM areas that these tracts structurally connect. Limited by the temporal resolution in this study (2 s), interpretation of the directionality of neural signal traveling in WM is unlikely. To avoid confusing readers, we removed the relevant text regarding signal traveling in this revision, but retained the concept of WM activations along relevant fiber tracts.

2101. Yali Huang, Stephen K. Bailey, Peiguang Wang, Laurie E. Cutting, John C. Gore, Zhaohua Ding, Voxel-wise detection of functional networks in white matter, *NeuroImage*, Volume 183, 2018, Pages 544-552.

2 You say that you can exclude that the stimulus-locked hemodynamic activity in white matter is due to the flux of blood to activated areas of grey matter. You bring some arguments towards this interpretation, but on the other hand we cannot ignore that the vascular system is unique and closed, and that blood arrives to different regions in the brain with some time delays.

Response: Given the speculation that BOLD signals in WM might be driven by residual effects from oxygenation changes in GM vasculature, we have paid considerable attention to the distributions of brain vasculature recently. From the architecture of arterial systems, it is known that WM has a dedicated blood supply system: it almost entirely receives blood supply from the medullary artery, which originates directly from the pial artery [2201]. This represents a distinct blood supply system from the GM. Meanwhile, there are two separate venous systems for GM and WM as well: a superficial venous system that drains deoxygenated blood in the cortex and

superficial WM into venous sinuses via cortical veins, and a deep venous system that drains deoxygenated blood into the subependymal veins [2202]. Distributions of veins and arteries in brain GM and WM are shown in Fig. R3.

Fig. R3. Brain artery supply (left, Takahashi, 2013) and brain venous architecture (right, Ruiz, 2009)

The differences in vasculature distributions between WM and GM are emphasized in our recent paper [2203] and is also briefly mentioned in this revised manuscript as follows:

“A common concern is that the observed WM activations may be spurious because of the residual effects from oxygenation changes in GM vasculature. For the arterial system, WM almost entirely receives blood supply from the medullary artery, which originates directly from the pial artery and does not give off branches in cortical areas [2201]. This represents a distinct blood supply system from the GM. For the venous system, unlike the cortices, whose venous drainage collects into the pial veins located at the surface of the brain, deep WM tracts are drained via the sub-ependymal veins near the lateral ventricles. As such, there are no vascular interactions between the two different tissue types, and the blood flow out of activated GM regions is unable to reach deep WM to modulate the signals therein (exceptions exist for developmental venous anomalies but these have a maximum incidence of only 2.6%).”

Therefore, it is unlikely that the fluctuations in WM BOLD signals are caused by sharing of vasculature with GM. We should further point out that vascular responses triggered by neural events are tightly restricted the activated site (within 100um) via an intricate control of the neurovascular unit, which ensures high site-specificity of neurovascular coupling [2204].

2201. Takahashi S, ed. Neurovascular Imaging: MRI & Microangiography. London: Springer; 2011.

2202. San Millán Ruíz, D. , Yilmaz, H. and Gailloud, P. (2009), Cerebral developmental venous anomalies: Current concepts. Ann Neurol., 66: 271-283.

2203. Yali Huang, Stephen K. Bailey, Peiguang Wang, Laurie E. Cutting, John C. Gore, Zhaohua Ding, Voxel-wise detection of functional networks in white matter, *NeuroImage*, Volume 183, 2018, Pages 544-552.

2204. Iadecola, C. Neurovascular regulation in the normal brain and in Alzheimer's disease. *Nat. Rev. Neurosci.* 5, 347 (2004).

3 Proper physiological control and correction would be in order (breath holding, measurement of cardiac pulse and respiration).

Response: Per your suggestion, the physiological sources of noise, primarily including cardiac pulsations and respiration-induced modulations, are removed using the CompCor approach [2301]. This approach regresses out nuisance variables that are extracted from a cerebrospinal fluid mask. The effectiveness of the CompCor has been assessed in previous report [2301] by comparing its measured power spectrum of physiological components with the sum of the cardiac and respiratory elements identified by photoplethysmograph sensor and respiratory belt. The result shows that CompCor is capable of capturing accurate physiological fluctuations and has the advantage of not requiring external monitors. With this additional preprocessing step, we updated all the relevant results, including Fig. 3 – Fig. 7, Supplementary Fig. 2- Fig 4 and Supplementary Table2, and added a brief description of CompCor to the Methods as follows:

“Confounding effects of physiological fluctuations, such as cardiac pulsations and respiration-induced modulations, on fMRI time-series were removed using a CompCor approach [2301]. Specifically, nuisance variables were derived from the anatomical noise ROI, identified within cerebrospinal fluid (CSF), and were subsequently regressed out from the BOLD time series.”

2301. Yashar Behzadi, Khaled Restom, Joy Liau, Thomas T. Liu, A component based noise correction method (CompCor) for BOLD and perfusion based fMRI, *NeuroImage*, Volume 37, Issue 1, 2007, Pages 90-101.

4 Time-locked activity is been found basically in all the brain if you look for it (<http://www.pnas.org/content/109/14/5487>, <https://academic.oup.com/cercor/article/25/12/4667/309934>). This makes less evident your choice of looking at specific regions and tracts, unless you don't validate somehow that the HRF in the tracts reflects the communication between the connected areas. This communication, if it exists, can be neuronal or physiological, in particular if we know already that we find HRF in white matter, and that your main contribution would be the causal role of the GM activation in the shape and existence of the HRF one. Your evidence is purely anecdotal, there is no validation of this hypothesis, nor a counterhypothesis. You should for example look at HRF everywhere in the brain, and see if the shape in WM is modulated by the activation of connected areas, possibly with different paradigms.

Response: The purpose of looking at specific tracts is not to investigate the communication between the connected GM areas, but to group WM voxels that are most likely evoked by the event. In our recent publication [2401], we investigated the BOLD signal evoked by visual stimuli using a voxel-wise manner throughout the brain based on a visual task. Our findings suggested that specific WM voxels are robustly activated, and that regional distributions of those

WM voxels are consistent with fiber pathways that bear functional relevance to visual activities. Meanwhile, we found that the time course of individual voxels often failed to exhibit clear patterns of responses to the visual stimuli due to low SNR and high variability. To increase the SNR in the current study, we therefore group WM voxels along individual tracts since they presumably share similar frequency spectra with the activated GM according to our previous findings. On the other hand, to eliminate the misleading statement, we removed the relevant text regarding signal traveling or communication between GM areas.

We recognize that brain responses to even a very simple stimulus may be wide-spread due to the integrative nature of brain functional organization. Meanwhile brain functions also exhibit the nature of spatial segregation, i.e., specialized neurons and brain areas are organized into distinct neuronal populations that form segregated cortical areas. Neural activity in such area is often detected on the basis of response strength, with an appropriate threshold to filter out the regions with weaker responses. This same notion is applicable to WM as well. In the response to comment 1 from review 3, we experimented detection of WM activations throughout the brain using new HRF models we derived from WM bundles. Setting a loose threshold ($p < 0.02$, uncorrected), we observed that a large portion of the activations detected with our WM-specific HRF are distributed in the anterior part of the body of corpus callosum and anterior coronal radiation. The distributions of these WM activations are consistent with neuroanatomy that there are fiber pathways that connect the activated frontal GM areas in the two hemispheres. This experiment also provides empirical support for the use of fiber tract that connects activated GM to derive the WM HRF on the same functional task.

2401. Yali Huang, Stephen K. Bailey, Peiguang Wang, Laurie E. Cutting, John C. Gore, Zhaohua Ding, Voxel-wise detection of functional networks in white matter, *NeuroImage*, Volume 183, 2018, Pages 544-552.

5 It has been recently confirmed that brain areas involved in task enlarge <https://www.biorxiv.org/content/early/2018/09/05/408658> . If this would be the case in your data, it could be that the grey matter would expand where the white matter is supposed to be in your mask.

Response: WM HRFs in this study are reported in 14 s time windows, and most HRFs reach the peaks before 10 s. Compared to the GM enlargement in previous report [2503] caused by the visual stimuli that lasted a few minutes, the GM enlargement, even if existing within the 14 s window, was triggered by a two-second colored picture, which we believe should be negligible. In addition, since the time courses are normalized to unit variance, the HRF magnitudes are relative to the baseline (the congruent events) during which the subjects watched a colored word picture as well. With visual effect controlled, the only cause of the enlargement would be the Stroop effect, which is instant in time and thus involves much less stimulation contrast than the picture viewing. The following paragraph summarizes our discussions in this regard, which can be added to the manuscript upon request of the reviewer.

“One may speculate that the observed BOLD signal waves in WM might be attributable to stimulus-induced morphological variations in GM. It has been reported that such morphological changes could occur after specific training for a few days or longer [2501, 2502]. Recently, it has been shown that rapid morphological enlargement of the visual cortex could be triggered by

picture viewing [2503]. Although the morphological changes were found to take place within a few minutes, the total amount of increase in morphometry was rather small (<1.5%), which was arguably consequential to increased blood flow nearby. In contrast, the stimulus paradigm in our work is impulse in nature, and presumably involves less stimulation contrast than the picture viewing, which in principle should not incur large expansion of GM that protrudes to the WM domain to induce measurable variations in the BOLD signals.”

2501. V. Kwok et al., Learning new color names produces rapid increase in gray matter in the intact adult human cortex. *Proc. Natl. Acad. Sci. U. S. A.* 108, 6686–6688 (2011).

2502. B. Draganski et al., Neuroplasticity: changes in grey matter induced by training. *Nature.* 427, 311–312 (2004).

2503. K.N.T. Mansson et al. Viewing pictures triggers rapid morphological enlargement in the human visual cortex

6 Some results and open maps exist from a data driven approach to retrieve a proxy of the hemodynamic response function at rest, including extended to white matter

<https://neurovault.org/collections/3584/>

<https://doi.org/10.6084/m9.figshare.7139702.v2>

You could possibly compare your results with those, to obtain a sort of a baseline.

Response: We applied the RS-HRF tool [2601] that created the proxy map to the resting-state data collected from the same group of subjects used in the current study. Fig. R4 displays HRFs of the ACC and four WM tracts that connect to it. The GM and WM exhibit HRF profiles with nearly identical magnitudes and shapes. This is not in keeping with the neurovascular mechanism in WM which was consistently demonstrated in various hypercapnia-based cerebrovascular reactivity (CVR) studies. In these studies, the BOLD signals reveal reduced and delayed CVR even in the subcortical WM. Thus it appears that the data-driven approach may not be robust enough in retrieving the HRF in WM.

Fig. R4. HRFs of ACC and four WM tracts retrieved by the RS-HRF tool

By modifying the RS-HRF codes, we were indeed able to retrieve the HRF from our Stroop data. We simply aligned the HRF to the onset of the Stroop event rather than searching for spikes above a pre-specified threshold, thus adapting the tool to stimulus-locked fMRI data. Fig. R5 displays HRFs of the ACC and four connecting tracts obtained from our task data by the

modified RS-HRF codes. While the profile of the GM HRF remains comparable to that in Fig. R4, attesting to the validity of our modification, the WM HRFs exhibit reduced magnitudes, delayed response and more pronounced initial dips, just like our findings in Fig. 3.

Fig. R5. HRFs of ACC and four WM tracts obtained from task data by using the modified RS-HRF codes

2601. G. Wu, G. Deshpande, S. Laureys and D. Marinazzo, "Retrieving the Hemodynamic Response Function in resting state fMRI: Methodology and application," 2015 37th Annual International Conference of the IEEE Engineering in Medicine and Biology Society (EMBC), Milan, 2015, pp. 6050-6053.

7. Why using the gamma/balloon models and not a Finite Impulse Response, or an approach based on physiological Gaussian priors
<https://www.biorxiv.org/content/early/2017/11/14/179838>?

Response: The purpose of using the balloon model in this work is to derive distinct sets of parameters which could be an extension to the canonical HRF, i.e., the one integrated in the well-known software SPM, which was modeled by gamma functions as well. Our experiments show that the fitted parameters of the HRFs in WM are drastically different from those of the canonical HRF, thus calling for significant changes to the standard gamma functions for more effective detection of neural activities in WM. The Finite Impulse Response and the model based on physiological Gaussian priors, however, are both nonparametric approaches. Though more flexible, they are not capable of producing parameters to represent the magnitude, shape, scale and time lag of either positive or negative pole, which are useful in understanding of the coupling between cerebral blood flow (CBF) and cerebral metabolic rate of oxygen (CMRO₂) in WM.

8. On page 7 you write: "Such connectivity patterns may provide the functional architecture for signals to be transmitted from one node to another among these activated clusters". This is a strange and recursive definition: there are coactivated areas, which come up with ICA or seed-based correlation, and we call them Functional Connectivity, as opposed to the Structural one. At these scales we don't need FC to infer SC, and there are many factors, including vascular and autonomic physiology (and movement) which contribute to the formation of FC patterns.

Response: We apologize for the misleading statements. We have deleted this statement, which is both controversial and redundant, from the discussions.

9 Figures: I appreciate that you show all the data points in the box plots. On the other hand p-values are an uniform distribution, no need of discretizing them with asterisks indicating different thresholds. Also, p-values should be accompanied with an effect size. If you have an hypothesis of the link between the activations in GM regions and connecting tracts, you should include it in the figure (the simplest model (i.e. no model) would imply just to link all the triplets of points for the same subject).

Response: Per your request, we removed the levels of statistical significance denoted by asterisks and added the effect size measured by Cohen's d value. We also detailed the tracts name in the figures to indicate a hypothesis of the link between the activations in GM regions and connecting tracts.

10 To summarize, it's likely that what you measure is actually hemodynamic response functions in white matter (the BOLD comes from vessels, mainly microvessels, see this recent study <https://www.ncbi.nlm.nih.gov/pubmed/29398359>), and there are vessels surrounding white matter (even though for this type of study I would have expected a more rigorous account of the role of respiration and cardiac phase). But you cannot exclude that what you measure comes (all or in part) from vessels in the grey matter.

What is not clear is that the white matter HRF that you find is a signature of a signal travelling between two activated areas, neither of any other sort of communication between them, apart from the mere existence of the white matter tracts.

Response: Please refer to our responses to your comments 2 and 3.

Reviewer #3 (Remarks to the Author):

Li et al. present an excellent study investigating white matter (WM) activation in functional MRI. I know the area well and was impressed with the quality of the experimental work and the drafted manuscript.

We greatly appreciate your positive evaluation of this work. Our point-by-point response follows each question below.

I Major suggestion: Recommend the authors add a figure show WM activation using the novel HRF model parameters to significantly improve impact.

Response: Per your suggestion, we experimented detection of WM activations using new HRF models we derived from WM bundles. Fig. R6 displays an example of detected activations of subject-averaged BOLD data using the canonic HRF model embedded in SPM (top row and middle row) and our custom regressor reconstructed by convolving incongruent stimuli with the fitted HRF for the tract Fron_L to Fron_R (bottom row). Compared to the canonic HRF, which detects activations mainly in GM, a large portion of the activations detected with our WM-specific HRF are distributed in the anterior part of the body of corpus callosum and anterior coronal radiation. The distributions of these WM activations are consistent with neuroanatomy that there are fiber pathways that connect the activated frontal GM areas in the two hemispheres. This figure is also included as Supplementary Fig. 5 in the manuscript.

Fig. R6. GM and WM activations detected using the canonic HRF model embedded in SPM (top row, $p < 0.05$ FWE corrected; middle row, $p < 0.02$ uncorrected) and a custom regressor reconstructed by convolving incongruent stimuli with the fitted HRF for the tract Fron_L to Fron_R (bottom row, $p < 0.02$ uncorrected).

2. Minor suggestions: One never wants to request additional citations related to our own work, but in this case I suggest that the authors consider adding these references:

1) In the initial WM fMRI reference (Lines 54-55) - Gawryluk et al., 2014 Frontiers Review paper - as this provides source material that will help substantiate WM HRF physiology interpretations and supporting literature.

2) Fraser et al. (2012) BMC Neuroscience and Yarkoni et al (2009) PloS One - as both of these specifically focus on WM HRF characteristics (in addition to Courtemanche et al.).

Response: These three contributions are very important and have now been cited in the manuscript.

3. Also, lines 166 and 167 are speculative and best moved to Discussion section rather than Results section.

Response: Thank you for this suggestion. This part has been moved to discussion section.

Reviewers' comments:

Reviewer #1 (Remarks to the Author):

I am generally satisfied with the responses of the authors to the issues raised. I have further comments.

First, regarding "proximal integration", one should certainly cite Itoh & Suzuki, JCBFM 2012; they coined the phrase, more or less. BOLD response modeling work based on this concept was in Kim & Ress, Neuroimage 2016.

Second, the logic regarding your choice to not use SMS is fine, but should also appear briefly in the Methods. I think it would also be helpful to briefly discuss how your limited spatial resolution might impact the results. In fact, your erosion procedure is not entirely clear. In what space did this erosion occur? The 1-mm-voxel high-resolution anatomy was my guess, but you should be specific. How much erosion was done on each pass? A single 1-mm erosion, given the fMRI spatial resolution, would be meaningless. That does not appear to be the case, fortunately, from Supp, Fig. 6, but please be more specific.

Reviewer #2 (Remarks to the Author):

Dear authors

thanks for the extensive and overall convincing rebuttal.

Concerning the possible confounding effect of blood travelling along white matter fibers on white matter activation, your explanations are not completely convincing, in particular considering this recent paper

"Rapid solution of the Bloch-Torrey equation in anisotropic tissue: Application to dynamic susceptibility contrast MRI of cerebral white matter"

<https://www.sciencedirect.com/science/article/pii/S1053811918320068>

showing that about half of the white matter blood resides in vessels that run in parallel with white matter tracts.

Also this paper

"Microvasculature of the human cerebral white matter: arteries of the deep white matter"

<https://onlinelibrary.wiley.com/doi/abs/10.1046/j.1440-1789.2003.00486.x>

shows that deep white matter arteries run straight through the white matter, but also pass through the cortex.

Taken together, these results speak to a possible confounder to pure white matter activation.

Daniele Marinazzo

Reviewer #3 (Remarks to the Author):

The authors have provided a figure addressing my primary concern. However, it is my view that

there remains three important revisions to this figure: 1) It should be in the primary manuscript, not a supplemental figure. If the goal is to demonstrate white matter activation, then this figure is critical to that end; 2) The Results and Discussion sections require additional revisions to describe this result in terms of hypotheses related to the white matter activation and corresponding functional neuroanatomy; and 3) Please include the $P < 0.05$ FWE corrected WM images to the figure for full transparency, even if they did not show white matter activation. It is also important to discuss the relative sub-threshold results if this was indeed the case, otherwise future studies will not necessarily appreciate the importance of applying GM-based thresholds relative to the requirement for specific methods to detect sub-threshold activity in WM.

Reviewer #1 (Remarks to the Author):

I am generally satisfied with the responses of the authors to the issues raised. I have further comments.

Response: We greatly appreciate your positive evaluation of the revised manuscript. Our point-by-point response follows each question below.

1 First, regarding "proximal integration", one should certainly cite Itoh & Suzuki, JCBFM 2012; they coined the phrase, more or less. BOLD response modeling work based on this concept was in Kim & Ress, Neuroimage 2016.

Response: These two important contributions have now been cited in the manuscript.

2 Second, the logic regarding your choice to not use SMS is fine, but should also appear briefly in the Methods. I think it would also be helpful to briefly discuss how your limited spatial resolution might impact the results.

Response: We added in the methods section our rationale for not using the SMS, as follows:

“The fMRI images were acquired from these subjects with TR = 2 s, TE = 35 ms, SENSE factor = 2, matrix size = 80×80 , FOV = 240×240 mm², 34 slices of 4 mm thickness with a 0.5 mm gap, and 200 dynamics. Simultaneous multi-slice (SMS) sequences, though yielding higher spatial resolution with comparable acquisition times, were not employed in the current study as our pilot data indicated that SMS tended to reduce SNR in BOLD signals as well as gray-white matter contrast, in addition to introducing greater geometric distortions.”

The limitation of the low spatial resolution of fMRI images collected in the current study is also discussed as follows:

“We recognize that, in this study, we acquired images with relatively high SNR and GM-WM contrast and low image distortion, but with a spatial resolution which could have introduced some partial volume effects. With the recent rapid evolution of SMS sequences and maturation of reliable imaging protocols, more favorable trade-offs between contrast/signal-to-noise ratio and spatial/temporal resolutions may be achieved.”

3 In fact, your erosion procedure is not entirely clear. In what space did this erosion occur? The 1-mm-voxel high-resolution anatomy was my guess, but you should be specific. How much erosion was done on each pass? A single 1-mm erosion, given the fMRI spatial resolution, would be meaningless. That does not appear to be the case, fortunately, from Supp, Fig. 6, but please be more specific.

Response: We detailed the erosion procedure in the methods as follows:

“Specifically, the Colin 27 Average Brain (CH2) WM template [1301] (MNI space, 1 mm voxel resolution) was taken as the initial input of the erosion operations. A sphere-shaped structuring element with radius of 4 mm was created to probe the input image and define the neighborhood used in the erosion of each pixel. The Level i mask was obtained by subtracting the output image from the input image during the i th iteration of erosions. Finally all the high-resolution masks were resliced to 3 mm voxel resolution to match the coordinates of the normalized fMRI images.”

[1301] Holmes CJ, Hoge R, Collins L, Woods R, Toga AW, Evans AC (1998): Enhancement of MR Images Using Registration for Signal Averaging. *J Comput Assist Tomogr* 22.

Reviewer #2 (Remarks to the Author):

Dear authors

thanks for the extensive and overall convincing rebuttal.

Response: We greatly appreciate your positive evaluation of the revised manuscript.

I Concerning the possible confounding effect of blood traveling along white matter fibers on white matter activation, your explanations are not completely convincing, in particular considering this recent paper, "Rapid solution of the Bloch-Torrey equation in anisotropic tissue Application to dynamic susceptibility contrast MRI of cerebral white matter", <https://www.sciencedirect.com/science/article/pii/S1053811918320068>, showing that about half of the white matter blood resides in vessels that run in parallel with white matter tracts.

Also this paper "Microvasculature of the human cerebral white matter: arteries of the deep white matter", <https://onlinelibrary.wiley.com/doi/abs/10.1046/j.1440-1789.2003.00486.x>, shows that deep white matter arteries run straight through the white matter, but also pass through the cortex. Taken together, these results speak to a possible confounder to pure white matter activation.

Response:

Thank you for suggesting these papers, which we did review before submitting our manuscript

Nonaka's paper [2101] reported that "the deep white matter arteries penetrated the cortex without branching, ran straight through the white matter, and concentrated ventriculopetally to the white matter around the angle of the lateral ventricle. As they penetrated the white matter, they displayed several long side branches with pedicels, dividing into many branches. These directed toward both the surface and deeper parts of the white matter, thereby supplying the surrounding white matter". As such, it would be highly unlikely that activations of the GM could have much impact on the "downstream" WM since the GM receives blood through a dedicated arterial system rather than through the arteries that penetrate the cortex to supply WM. In fact, it has been experimentally demonstrated in a rat model that increases in cerebral blood flow in response to neural activity are tightly restricted to the site of activation, with submillimeter precision [2102, 2103]. In addition, Nonaka's paper also pointed out that "no branches from the lenticulo-striate arteries were seen to irrigate the cerebral white matter." Therefore, there is little possibility either that WM activation is confounded by the vasculature of the subcortical nuclei which receive blood supplies from lenticulo-striate arteries.

Doucette's paper [2104] reported that about half of the WM blood resides in vessels that run in parallel with WM tracts. This raises the possibility that the signal fluctuations observed in WM might not be purely BOLD since factors other than the level of oxygenation could modulate local magnetic susceptibility as well, so that the progressively increased time of delay in some WM tracts could be partly attributable to the susceptibility effect in the parallel vessels. However, as the confounding factors tend to shorten T_2 relaxation times and thus decrease the magnitude of BOLD signals, it is unlikely that these confounders contribute to the positive peak of the BOLD signals we observed, though in principle they could increase the initial negative dips.

The above is incorporated into relevant discussions as follows:

“A common concern is that the observed WM activations may be spurious because of residual effects from oxygenation changes in GM vasculature. Anatomically, WM receives blood supply almost entirely from the medullary artery³⁴, which originates directly from the pial artery and does not give off branches to cortical areas³⁵. Although the arteries supplying WM pass through cortical GM, it seems highly unlikely that BOLD signal changes produced by activation of the GM could have much impact on “downstream” WM because the GM receives blood through a separate, dedicated arterial system rather than through the arteries that penetrate the cortex to supply WM. In fact, it has been experimentally demonstrated in a rat model that increases in cerebral blood flow in response to neural activity are tightly restricted to the site of activation, with sub-millimeter precision^{36,37}. With respect to the venous system, a large activated GM cluster may generate oxygenation changes along draining veins up to several mm beyond the edge of the activated cluster³⁸. However, unlike cortical areas, whose venous drainage collects into the pial veins located at the surface of the brain, deep WM tracts are drained via the subependymal veins, which are close to the lateral ventricles³⁹. As such, there are no vascular interactions between the two different tissue types, and the blood flow out of activated GM regions is unable to reach deep WM to modulate the signals therein (exceptions exist for cases of developmental venous anomalies but these have an incidence only up to 2.6%⁴⁰). Another potential concern is that the signal fluctuations we observed in WM might not be purely BOLD in origin because factors other than the level of oxygenation could modulate local magnetic susceptibility as well. For example, the progressively increased time delay of signal changes in some WM tracts could be partly attributable to the influence of susceptibility changes in parallel vessels, in which half of the WM blood resides⁴¹. However, such potential confounding factors should tend to shorten transverse relaxation times and thus would decrease the magnitude of MRI signals, so it is unlikely that such factors account for the positive peaks of the BOLD signals we observed, though in principle they could contribute to the initial negative dips.”

[2101] Nonaka, H, Akima, M, Hatori, T, Nagayama, T, Zhang, Z and Ihara, F (2003), Microvasculature of the human cerebral white matter: Arteries of the deep white matter. *Neuropathology*, 23: 111-118.

[2102] Iadecola C (2004): Neurovascular regulation in the normal brain and in Alzheimer’s disease. *Nat Rev Neurosci* 5:347–360.

[2103] Chaigneau E, Oheim M, Audinat E, Charpak S (2003): Two-photon imaging of capillary blood flow in olfactory bulb glomeruli. *Proc Natl Acad Sci* 100:13081 LP-13086.

[2104] Jonathan Doucette, Luxi Wei, Eneidino Hernández-Torres, Christian Kames, Nils D. Forkert, Rasmus Aamand, Torben E. Lund, Brian Hansen, Alexander Rauscher, Rapid solution of the Bloch-Torrey equation in anisotropic tissue: Application to dynamic susceptibility contrast MRI of cerebral white matter, *NeuroImage*, Volume 185, 2019, Pages 198-207.

Reviewer #3 (Remarks to the Author):

The authors have provided a figure addressing my primary concern. However, it is my view that there remains three important revisions to this figure.

Response: We greatly appreciate your positive evaluation of the revised manuscript. Our point-by-point response follows each question below.

1 It should be in the primary manuscript, not a supplemental figure. If the goal is to demonstrate white matter activation, then this figure is critical to that end.

Response: Per your suggestion, this figure has been moved to the primary manuscript (Fig. 8).

2 The Results and Discussion sections require additional revisions to describe this result in terms of hypotheses related to the white matter activation and corresponding functional neuroanatomy.

Response: Per your suggestion, we add more detailed description of this result as follows:

“To test the relevance of the derived HRFs, we applied them to detect WM activations in selected WM bundles. Fig. 8 displays an example of detected activations of subject-averaged BOLD data using both the canonical HRF model embedded in SPM (first row and second row) and our customized regressor reconstructed by convolving the incongruent stimuli time courses with the fitted HRF for the tract Fron_L to Fron_R (third row and fourth row). The WM activations (fourth row) are distributed in the anterior part of the body of the corpus callosum and anterior coronal radiation and are consistent with the known presence of fiber pathways that connect the activated frontal GM areas in the two hemispheres. On the contrary, these activations did not show up in the activation map based on the canonical HRF even with a loose threshold ($P < 0.02$, uncorrected). When tightening the threshold ($P < 0.05$, FWE corrected), the activation in GM trends to converge on the cluster center while the activation in WM is no longer detectable.”

We also discuss this result as follows:

“We have demonstrated that the activation of deep WM was not detectable when using the canonical HRF for GM even with loose constraints, but could be detected with our customized regressor derived from a tract-specific HRF. However, it is also worth emphasizing that WM activity will often not show up if the same criteria (threshold and cluster size) are applied as are used for GM BOLD effects because they are substantially weaker.”

3 Please include the $P < 0.05$ FWE corrected WM images to the figure for full transparency, even if they did not show white matter activation.

Response: Per your suggestion, the WM activation map ($P < 0.05$, FWE corrected) has been added to Figure 8.

4 It is also important to discuss the relative sub-threshold results if this was indeed the case, otherwise future studies will not necessarily appreciate the importance of applying GM-based thresholds relative to the requirement for specific methods to detect sub-threshold activity in WM.

Response: Please refer to our response to your comment 2.

REVIEWERS' COMMENTS:

Reviewer #2 (Remarks to the Author):

Dear authors

thanks for this reply.